# The Development and Application of the Urban Flood Risk Assessment Model for Reflecting upon Urban Planning Elements

**Kiyong Park [1] and Man-Hyung Lee [2,\*]**

[1] Dept. of Disaster Prevention, Chungbuk National University, 1 Chungdae-ro, Seowon-gu, Cheongju, Chungbuk 28644, Korea; pky3489@chungbuk.ac.kr

[2] Dept. of Urban Engineering, Chungbuk National University, 1 Chungdae-ro, Seowon-gu, Cheongju, Chungbuk 28644, Korea

\* Correspondence: manlee@chungbuk.ac.kr; Tel.: +82-10-7623-2369

**Abstract:** As a city develops and expands, it is likely confronted with a variety of environmental problems. Although the impact of climate change on people has continuously increased in the past, great numbers of natural disasters in urban areas have become varied in terms of form. Among these urban disasters, urban flooding is the most frequent type, and this study focuses on urban flooding. In cities, the population and major facilities are concentrated, and to examine flooding issues in these urban areas, different levels of flooding risk are classified on 100 m × 100 m geographic grids to maximize the spatial efficiency during the flooding events and to minimize the following flooding damage. In this analysis, vulnerability and exposure tests are adopted to analyze urban flooding risks. The first method is based on land-use planning, and the building-to-land ratio. Using fuzzy approaches, the tests focus on risks. However, the latter method using the HEC-Ras model examines factors such as topology and precipitation volume. By mapping the classification of land-use and flooding, the risk of urban flooding is evaluated by grade-scales: green, yellow, orange, and red zones. There are two key findings and theoretical contributions of this study. First, the areas with a high flood risk are mainly restricted to central commercial areas where the main urban functions are concentrated. Additionally, the development density and urbanization are relatively high in these areas, in addition to the old center of urban areas. In the case of Changwon City, Euichang-gu and Seongsan-gu have increased the flood risk because of the high property value of commercial areas and high building density in these regions. Thus, land-use planning of these districts should be designed to reflect upon the different levels of flood risks, in addition to the preparation of anti-disaster facilities to mitigate flood damages in high flood risk areas. Urban flood risk analysis for individual land use districts would facilitate urban planners and managers to prioritize the areas with a high flood risk and to prepare responding preventive measures for more efficient flood management.

**Keywords:** climate change; urban flood risk; flood damage; urban disaster; land use

## 1. Introduction

The degree and scale of flood hazards have massively increased with the changing climate in the last decades. The larger-scale flash floods than in the past have brought fast-moving and rapid-rising water with force, resulting in tremendous life and property losses, as well as social disruption worldwide [1].

Floods are natural processes in river systems [2,3]. However, humans have occupied and urbanized floodplains for their urbanization attractiveness due to their planar morphology and water

availability [4,5]. The latter has translated into the growth of flood risk zones for human settlements and infrastructure due to a greater concentration of people and structures [6,7]. Recent urban growth has not taken the space that rivers require to temporarily store flows during floods into consideration [4]. Global efforts have focused more on implementing flood control infrastructure, such as dikes, dams, and channelization, but despite these efforts, modern cities still remain vulnerable to flood risk [8].

Furthermore, the potential for flood casualties and damages is also increasing in many regions due to the social and economic development, which implies pressure on land-use, e.g., through urbanization. Flood hazard is expected to increase in frequency and severity, through the impacts of global change on climate, resulting in severe weather in the form of heavy rains and river discharge conditions [9].

More cities are becoming hotspots for risk and disaster [10], mainly as a result of rapid urbanization, population growth, and the impacts of climate change [11].

Throughout the world, cities have been affected by the increasing impacts of floods. In the period 1998–2008, more than 2900 events were registered [12]. Recent estimates indicate that urban zones exposed to flooding will increase 2.7 times by the year 2030 [13]. Globally, the previewed scenario demonstrates an increase in the frequency and magnitude of floods due to the changes in precipitation patterns resulting from climate change and accelerated urban expansion [14–17]. It is estimated that by 2050, 70% of the world's population will be concentrated in urban areas [13,15,18].

Countermeasures to urban flooding should be considered in long-term perspectives because the impacts of climate change are unpredictable and complex [19].

The development of appropriate flood risk management strategies for flooding should be considered in long-term perspectives (e.g., expected future rainfall amounts, although climate change impacts can be unpredictable and complex) and should focus on increasing an area's resilience to flooding. Recently, as rainfall has become more concentrated over short periods of time, substantial amounts of damage due to pluvial flooding have occurred in urban areas. This includes damage of social infrastructures, as well as losses of human life and properties. Conventionally, storm water management has been focused on drainage systems via underground pipes. However, these conventional approaches (structural measures) have had problems in many cases because they were designed based on historical events. This makes it difficult to deal with extreme rainfall events that exceed the designated capacity. The frequency of extreme rainfall events is expected to increase with projected climate change, which may cause conventional stormwater management systems to be exceeded [20].

While information on the distribution of flood hazard at a national to global scale is extremely valuable, knowledge about how it is distributed with reference to the population and built assets is critical to water resource managers, city planners, and policy-makers [21].

In the past, natural disasters were accepted as an unavoidable calamity, and the emphasis was placed on how to cope with the aftermath of the disaster, rather than how to prevent it. However, the prevention measure shall shift its focus to preemptive prevention from a holistic point of view considering the large scale, unexpectedness, and complexity of the disaster. This study suggests the development and application of the urban flood risk assessment model to reflect upon urban planning elements (land use, building characteristics). In this respect, measuring in terms of urban planning is the most effective preemptive measure to reduce damage from natural disasters, and it focuses on the idea that the development shall be inhibited in the area that is vulnerable to flood damage and carries a high risk from the stage of land utilization planning in order to minimize the damage. In addition, an urban utilization plan may be an important policy measure in achieving the goal of keeping the city safe by restricting development in areas vulnerable to natural disaster. It is important to note that the impact of urban land use on disaster damages may escalate when unplanned and thoughtless developments prevail without the construction of appropriate infrastructure facilities and that this may become a reality while damages are currently increasing in urban areas due to reckless land use. Reducing the area for urban land use would naturally reduce the resilience of disaster damages. However, it would be difficult to expect such a reduction in the

area for urban land use as there has not been any reduction recently. Therefore, the purpose of this study is to analyze the flood risk by concentrating on land use and building characteristics reflecting on urban planning elements. The flood risk is classified to maximize the spatial efficiency during the flooding events and to minimize the following flooding damage. The paper concludes by stressing the necessity of land use planning measures to prevent flooding. It is possible to classify flood risk areas according to the use of districts and administrative districts, evaluate priorities, and enable efficient management by selecting areas with a high flood risk.

## 2. Methods

In Section 2.1 and Section 2.2, the theoretical perspectives are presented. Following this, Section 2.3 explores the relationship between urban space and flooding. In Section 2.4, the overall process of this study is described. Furthermore, in Section 2.5, the evaluation item of the analysis is presented. In Section 2.6, the fuzzy classification is introduced. Finally, in Section 2.7, the background of the selection of the analysis area is given.

### 2.1. Vulnerability and Exposure of Urban Flood

The risk of disaster is discontinuous and local. The only way to reduce the risk is to reduce the vulnerability within the system. The risk that places stress on the system is purely natural and is caused by variations outside of the system, and there is nothing that people can do to reduce the risk itself. However, human beings can only reduce the vulnerability to natural disasters by changing social systems or social infrastructure.

The concept of vulnerability defined by the IPCC is based on the view that combines vulnerability as the result of external stress and vulnerability as an internal state of the system. Vulnerability is the state of being easily affected by adverse impacts of climate change, including climate variations and extreme events, or the degree of inability to cope with it, and is the function of the characteristics, scale, and speed of climate variations to which a system is exposed and the sensitivity and susceptibility of the system [22,23]. In this respect, vulnerability should be regarded as potential exposure to damage rather than an estimation of damage due to a sudden change in climate or stress based on probability.

In the 8th IHP Strategic Plan of UNESCO, the specific concept of vulnerability was defined as in Figure 1, based on the disaster characteristics-damage relationship, by focusing on the idea that there is a difference in the scale of damage due to vulnerability, even for natural disasters of the same intensity. As explained above, the conceptual definition of vulnerability may be slightly different. However, it can be summarized that when the impact of natural disasters is great, the vulnerability of a system is considered high if it has a small capacity to cope with it. On the contrary, the capacity to cope with disaster can be high, even when the impact of natural disasters is high. This system can have the opportunity for development while coping with disasters appropriately. If the system has a small capacity to cope and the natural disaster has a small impact, the system may still have residual risks. If the impact is small and the system has a great capacity to cope, the system may promote sustainable development.

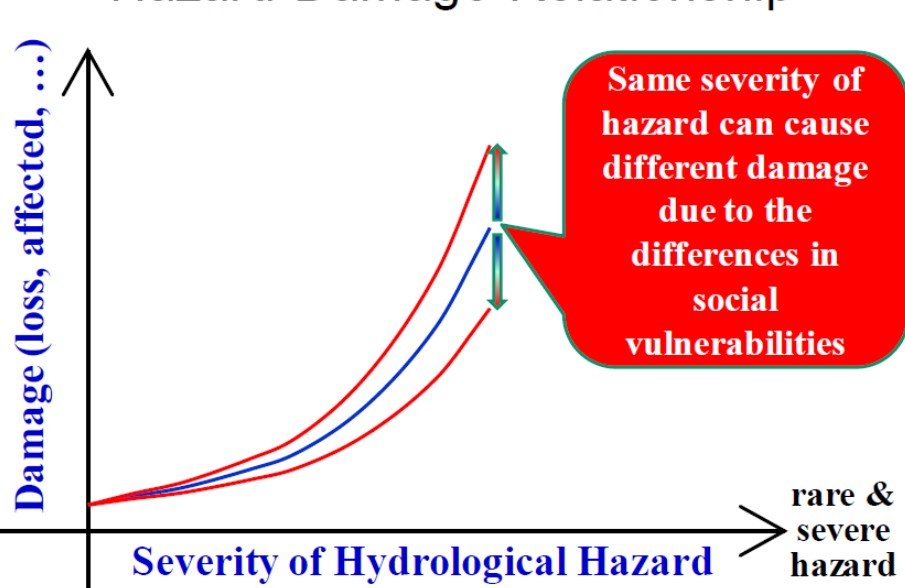

**Figure 1.** Relationship between the damage caused by hydrological extremes and the severity of the event [24].

Degree of exposure has been defined as the person, property, system, or other components in dangerous areas affected by a potential loss and can be measured by the number of people and assets in the area. To quantitatively estimate the risk involved with the hazard, the exposure can be combined with specific vulnerabilities of the elements exposed to a specific risk.

*2.2. Risk of Urban Flood*

The risk can be defined as the possibility that the loss will be greater than what was generally expected. In other words, risk is a concept of probability based on possibility, but it can be re-defined as the difference between expectation and reality considering the amount of loss [25].

According to ISO 31000, risk is defined as the "effect of uncertainty toward an object" or "the combination of the probability that an event would occur and the outcome of the event." The probability that an event will occur is related to the source of the disaster and its properties, and the outcome is related to vulnerability, which influences the scale of damage and the capacity to reduce damage.

There are still many cases where the meaning of risk is not clearly defined and is used confusedly to mean the degree of risk, hazardousness, or vulnerability. In this study, the components of risk considered are riskiness, vulnerability, and exposure, and also include the ability to adapt to risks.

*2.3. Relationship between Urban Space and Flooding*

Urban spaces carry a high risk of massive and complex human and property damages as they are quite vulnerable to disasters due to the concentration and densification of the population and facilities, the increased interdependence of various urban facilities and activities, the development of lowlands and slopes, and the development of underground space [26].

According to the fifth report of the IPCC, the artificial activity of human beings accounts for an absolute ratio of 95% of the cause of climate change. Paradoxically, however, human beings are also the ones that are most affected by diversified and escalated damages from various disasters caused by climate change. The city is a space where such human activities are concentrated, and the city, as well as the environment that surrounds people, influence and are influenced by climate change. Recent studies on disasters show that the impact of a natural disaster is caused by the interaction

between natural phenomena and people [27]. Eventually, the spaces that explain this phenomenon gather and form a city.

Urbanization and climate change have direct and indirect impacts on disasters or cause such disasters within urban spaces. Urbanization and climate change, which have great relevance across all areas of the urban space, are considered to be important paradigms related to natural disasters. If urbanization was the global phenomenon that drew the attention of the whole world in the 1900s, the international issue of the 2000s is climate change. The most common opinion among experts is that the influence of climate change will appear more prominently in society along with the acceleration of urbanization [28]. They argue that climate change is increasing natural disaster damage around the world, and the trend of urbanization is escalating damages [22].

The urbanization rate around the world expanded from 23.8% in 1950 to 50% in 2010. Urbanization is a population-concentrating phenomenon that appears after the concentration of various facilities and functions, and it is accompanied by various environmental problems [29]. The KOSIS (Korean Statistical Information Service) data also predicted that the urbanization rate will reach 60% or over as of 2030. Although the increase rate of urbanization has decreased after exceeding 50% in 2010, the rate of urbanization itself is expected to increase continuously. Assuming that the current environment remains the same, it is clear that the damage resulting from natural disasters will increase [30].

Urbanization is concentrating the limited urban space with artificial factors such as population, industries, and facilities, and increasing urban spaces vulnerable to natural disasters through indiscreet land use. In particular, unpredictable natural disasters due to climate change not only increase the risk even further in urban spaces with high vulnerability and urban spaces with less recovering capabilities but also form a vicious cycle of causing damage from natural disasters.

*2.4. Methods of Risk Analysis*

The basic ideas and concepts are shown above. The vulnerability analysis was carried out based on urban space characteristics and building characteristics, including non-structural characteristics, by setting 100 m × 100 m geographic grids as the evaluation units alongside urban flooding analysis based on environmental factors such as topographical characteristics and rainfall characteristics. The urban flood risk was assessed and analyzed based on these results and using the risk mapping technique. Concepts and processes important to their construction are described herein and in Figure 2.

1) Vulnerability analysis: Vulnerability is the result of external stress, which implies an internal condition of urban areas. It is the degree of socio-economic damage to the urban space (land use) and building characteristics (land price, floor area ratio, underground area, decline of building, material of building) during flooding.

2) Exposure (Hazard) analysis: An empirical analysis is of the flood depth, and flooding area was conducted considering the environmental factors, such as rainfall and topography. It is frequently referred to as an "inundation trace map", which is the result of an urban flooding map.

3) Flood risk analysis: This established the flood risk evaluation model analysis system by reflecting on the land use and building characteristics which are included in the non-structural measures to perform evaluation and analysis. Flood risk was calculated by deriving a value from each grid and overlapping it with the map by considering the vulnerability data, which classified the risk in each grid unit, and the analysis data for flooded areas, such as disaster characteristics, were employed as a function.

4) Urban flood risk assessment: Areas with high and low risks were determined by objective and scientific methods based on the urban flood risk assessment. Finally, the intent was to establish a long-term measure by setting the flood risk area based on the results analyzed by these objective and scientific methods.

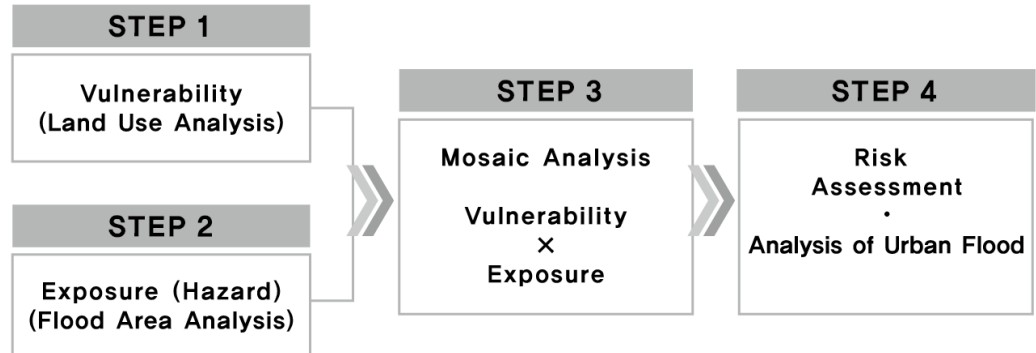

**Figure 2.** Analysis framework of urban flood risk.

*2.5. Evaluation Item*

Objective indicators and values were calculated to analyze the risk of flooding. Each indicator was selected based on the following standards.

The land price for each use district provided by the Korea Appraisal Board is considered an important indicator of flood damage because property damage represents the major loss in the case of a flood [31]. As the Ministry of Construction and Transportation established long-term water resource use planning in 2001, local characteristics of water resources were carefully examined to identify investment priority in watershed planning and development. As a result, Potential Flood Damage (PFD) was estimated, for which the property value plays an important role [32]. In the research of Han [33], the correlation between various indicators of flood vulnerability and the damage costs for flooding from 1971 and 2000 in different watersheds was analyzed. As a result, the property density had a significant correlation with the damage cost of flooding, which was a result attributed to the estimation method of flood damage cost largely reflecting the land price [33].

The underground area index is directly related to the flood reference system to prevent building submergence within the flood-water disaster prevention criterion for buildings [34]. In a study of Jang [35], the vulnerability of buildings due to flooding was found to be closely related to the presence and size of underground space. The top 10 items with the greatest damage were located in places where there were significant portions of underground space, while the bottom 10 items had no underground space. Therefore, the size of underground space was important for determining the vulnerability of buildings. Additionally, it is considered that rainwater adversely affects the building structures by moisture, warping, cracks, and corrosion when rainwater fills the underground space or penetrates through the wall [35].

The floor area ratio is a concept of disaster that collectively refers to the case where urban spaces lose their function and damages are multiplied due to a high density. Because of the high utilization of urban space, complex factors such as underground space and humans instigate unpredictable large-scaled disasters. In the United States, the Community Rating System (CRS), which was established in 1990 to operate the flood insurance system, has been employed to evaluate the building height issued by the Federal Emergency Management Agency (FEMA) for 19 items. Since the damage is expected to vary depending on the area and density of the building, the indicators are selected by considering urban functional damage [31].

A decline of building shows that the incidence of accidents is frequent as the durability reaches its limit. Moreover, the development-centered paradigm of urban planning has escalated an event to a series of failures, resulting in a disaster. Therefore, urban disasters have a high potential to result in a collective paralysis of urban functions [32].

The material of buildings is a robustness-related index. Unanwa et al. (2000) stated that property values of the building and exterior wall types were set. In the case of building interiors, it has been shown that many damages occur during flooding due to the heavy use of wood [36].

### 2.6. Fuzzy Classification

Fuzzy set theory has been developed and extensively applied since 1965 [37]. It was designed to supplement the interpretation of linguistic or measured uncertainties for real-world random phenomena. These uncertainties could originate with non-statistical characteristics in nature that refer to the absence of sharp boundaries in information. However, the main source of uncertainties involved in a large-scale complex decision-making process may be properly described via fuzzy membership functions [38].

Fuzzy classification is, alongside neural networks [39] and probabilistic approaches [40], a very powerful soft classifier. As an expert system for classification [41], it takes into account uncertainty in sensor measurements, parameter variations due to limited sensor calibration, vague (linguistic) class descriptions, and class mixtures due to a limited resolution. Fuzzy classification consists of an n-dimensional tuple of membership degrees, which describes the degree of class assignment μ of the considered object obj to the n considered classes.

$$f_{class,obj} = \left[ \mu_{class_1}(obj), \mu_{class\_2}(obj), \dots \mu_{class\_n}(obj) \right] \tag{1}$$

Crisp classification would only provide information on which membership degree is the highest, whereas this tuple contains all information about the overall reliability, stability, and class mixture. Fuzzy classification requires a complete fuzzy system, consisting of the fuzzification of input variables, resulting in fuzzy sets, fuzzy logic combinations of these fuzzy sets, and defuzzification of the fuzzy classification result to get the common crisp classification for map production. Fuzzy logic is a multi-valued logic quantifying uncertain statements. The basic idea is to replace the two Boolean logical statements "true" and "false" by the continuous range from 0 to 1, where 0 means "false" and one means "true", and all values between 0 and 1 represent a transition between true and false. Avoiding arbitrary sharp thresholds, fuzzy logic is able to approximate real-world complexity much better than the simplifying Boolean systems do. Fuzzy logic can model imprecise human thinking and can represent linguistic rules. Hence, fuzzy classification systems are well-suited to handle most sources of vagueness in remote sensing information extraction. The mentioned parameter and model uncertainties are considered by fuzzy sets, which are defined by membership functions. Fuzzy systems consist of three main steps, including fuzzification and the combination of fuzzy sets [42].

In the flood damage risk classification analysis, it is very difficult to quantify and compare the flood damage in commercial and residential areas when the same area is submerged. In other words, it is ambiguous to express the level of flood damage as numerical values. Therefore, to overcome the linguistic ambiguity for decision-making in previous studies and to analyze complex relationships between different indicators and indices, the fuzzy logic method is adopted to perform a more objective analysis of flood risk by deriving quantitative and accurate indicators.

### 2.7. Analysis Area

The distribution of flood damage in Korea analyzed by the Korea Ministry of Land, Infrastructure, and Transport is shown in Figure 3 [37], and Changwon City is selected as the target area due to its advantages of data construction because it includes areas with both high and low flood damage [31].

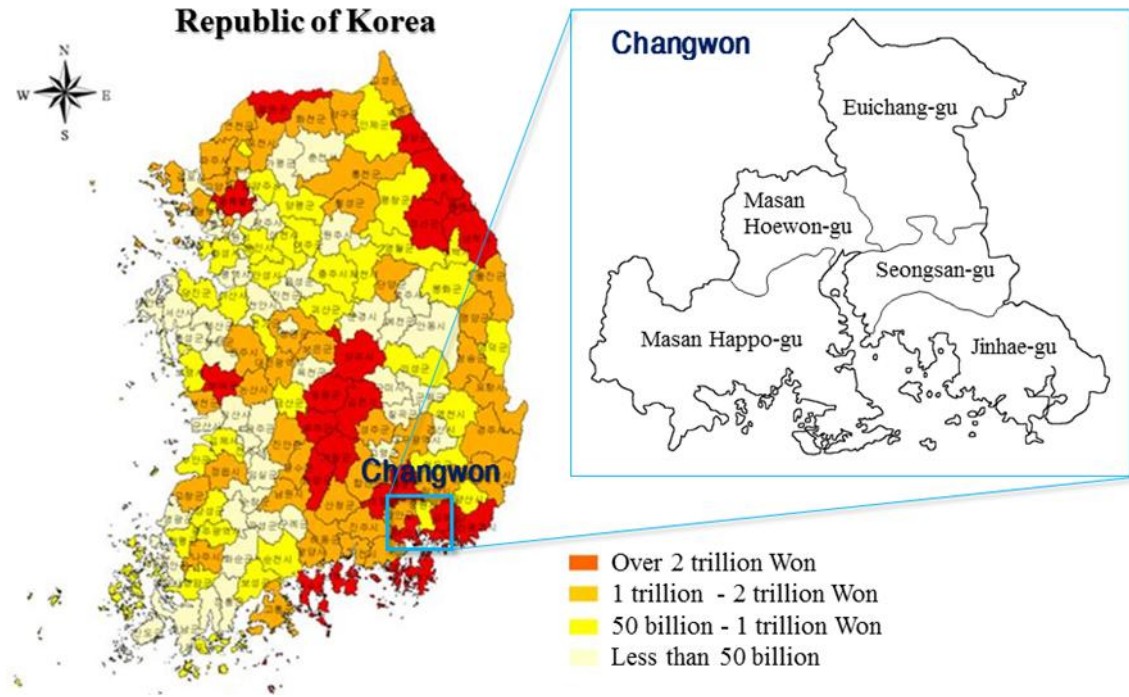

**Figure 3.** Distribution of national flood damage in Korea [43].

Changwon City has become one of the first successful administrative integration models in the nation as it integrated Changwon, Masan, and Jinhae cities in July 2010, and it is becoming the first growth base of the Southeastern Greater Economic Zone in Korea. The administrative district consists of the five districts of Masan Happo-gu, Masan Hoewon-gu, Seongsan-gu, Euichang-gu, and Jinhae-gu, 351 legal smaller districts, and 62 administrative smaller districts. The administrative area of Changwon City is 746.58 km² [44].

As a result of examining the land use situation from 1975 to 2007, using the land cover map of Changwon City, the urbanization rate greatly increased from 3.95% in the 1990s to 13.61% in recent years. In addition, a countermeasure to flood damage in Changwon city is needed because the urbanization rate is planned to increase from 14.13% in 2020 to 15.72% in 2025, according to the step-by-step development plan of Changwon City Basic Plan 2025 [44]. In response, the risk according to flood damage was analyzed in Changwon City.

## 3. Results and Discussion

### 3.1. Analysis of Vulnerability

To select the indicators capable of assessing the flood vulnerability to climate change, indicators related to land use and building in the non-structural aspect that were proposed in previous research were selected, and the final indicators were selected through content validity analysis. To calculate the quotient, the fuzzy methodology was used to derive the position function, and the fuzzy inference rules were established and implemented to analyze the vulnerability of the districts of Changwon, as seen in Figure 4. As all the spaces, buildings, and facilities in the city are basically the subject of disaster prevention, the characteristics of the areas were analyzed in 100 m × 100 m geographic grids, which maintains the livelihood of the citizens and the urban function.

There were a total of 177,193 cases of the officially assessed land prices for each lot in Changwon city, and the data were constructed using the individual officially assessed land prices provided by the Korea Appraisal Board. There were a total of 66,269 cases of the floor area ratio, 15,021 cases of underground area, 97,346 construction declines of buildings, and 69,932 cases of building materials according to the building register. The map of Changwon City was divided into 77,737 cells of 100 m × 100 m geographic grids, and maximum and minimum values, as well as the average value, were

standardized for the officially assessed land price, floor area ratio, underground area, decline of building, and building material indices, and fuzzy analysis was conducted to evaluate the vulnerability. As a result of the fuzzy analysis, the index with the highest vulnerability to flooding was land price, followed by underground area, floor area ratio, decline of building, and material of building. The implication of the result is that the land price and the underground space are direct indicators of property damage and physical damage in the case of flooding, whereas the floor area ratio is directly related to the only corresponding floor and indirectly related to other floors for its inconvenience in urban functional aspects. Furthermore, it is shown that decline of building and material of building are robustness-related indexes, which is less important than other indicators.

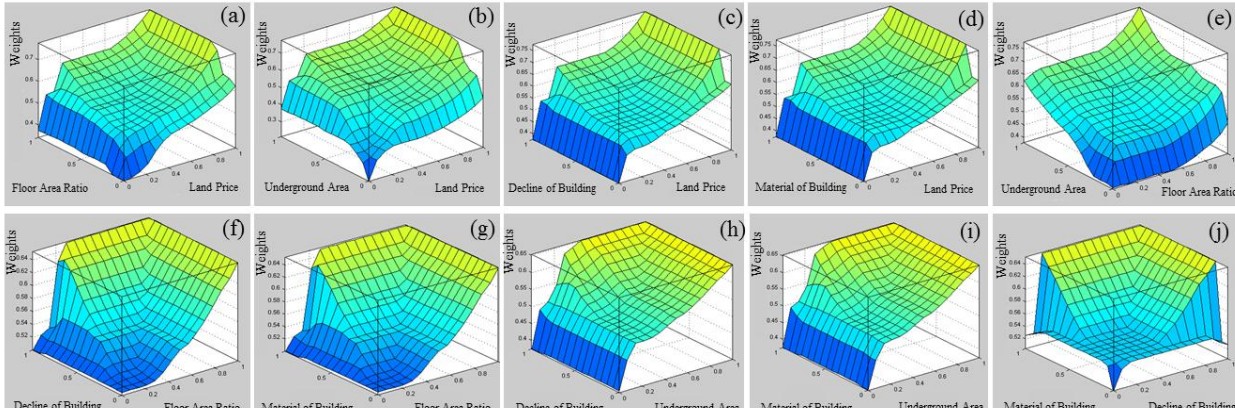

**Figure 4.** Flood damage scheme: (**a**) land price—floor area ratio, (**b**) land price—underground area, (**c**) land price—decline of building, (**d**) land price—material of building, (**e**) floor area ratio—underground area, (**f**) floor area ratio—decline of building, (**g**) floor area ratio—material of building, (**h**) underground area—decline of building, (**i**) underground area—material of building, and (**j**) decline of building—material of building.

A higher fuzzy score indicates a higher degree of flood risk (greater flood damage), and a lower fuzzy score indicates a lower degree of flood risk (no damage). The fuzzy values for 100 m × 100 m geographic grids were derived from fuzzy analysis through standardized values of each indicator for officially assessed land price, floor area ratio, underground area, decline of building, and building materials, listed in Table 1.

**Table 1.** Fuzzy values for 100 m × 100 m geographic grids.

| | Standard Values | | | | | Fuzzy |
|---|---|---|---|---|---|---|
| | Land Price | Floor Area Ratio | Underground Area | Decline of Building | Material of Building | |
| 1 | 0.5408 | 0.7624 | 0.0529 | 0.0356 | 0.0272 | 0.6150 |
| 2 | 0.6134 | 0.1774 | 0.0107 | 0.0910 | 0.0243 | 0.5552 |
| 3 | 0.6134 | 0.5305 | 0.0055 | 0.0742 | 0.0063 | 0.5488 |
| 4 | 0.6396 | 0.5612 | 0.0049 | 0.0425 | 0.0066 | 0.5475 |
| 5 | 0.5185 | 0.5996 | 0.0107 | 0.0430 | 0.0082 | 0.5454 |
| 6 | 0.1084 | 0.0001 | 0.0001 | 0.0001 | 0.0001 | 0.0950 |
| 7 | 0.1092 | 0.0925 | 0.0001 | 0.0356 | 0.0142 | 0.2200 |
| 8 | 0.0900 | 0.0001 | 0.0001 | 0.0341 | 0.0079 | 0.0950 |
| 9 | 0.1065 | 0.1136 | 0.0272 | 0.0346 | 0.0126 | 0.2272 |
| 10 | 0.1028 | 0.1067 | 0.0001 | 0.0338 | 0.0081 | 0.2200 |
| 11 | 0.1028 | 0.1335 | 0.0001 | 0.0326 | 0.0078 | 0.2500 |
| 77731 | 0.1779 | 0.0001 | 0.0001 | 0.0001 | 0.0001 | 0.2200 |
| 77732 | 0.2473 | 0.0967 | 0.0014 | 0.0059 | 0.0028 | 0.2377 |
| 77733 | 0.173 | 0.2219 | 0.0128 | 0.0089 | 0.0059 | 0.3388 |
| 77734 | 0.173 | 0.0001 | 0.0001 | 0.0001 | 0.0001 | 0.2200 |
| 77735 | 0.0979 | 0.0924 | 0.0023 | 0.0831 | 0.0028 | 0.2200 |

| 77736 | 0.0124 | 0.0473 | - | 0.0593 | 0.0042 | 0.0950 |
| 77737 | 0.0376 | - | - | - | - | 0.0950 |

Based on the fuzzy analysis value of each indicator, 100 m × 100 m geographic grids were prioritized for flood damages. When the same area was flooded, the area with the most flood damage in a social and economic sense, which is the area with high vulnerability, was analyzed as the area with the highest development density in Changwon centering on the central commercial area. In addition, the areas that played key functions in the city in the past as the old town center, although currently declining, showed high vulnerability.

Once the vulnerability grade for land use has been determined based on standardization of the fuzzy score, it is possible to classify based on the level of risk. The standard color was determined by designating the area with the highest vulnerability as the red zone, the area with high vulnerability as the orange zone, the area with intermediate vulnerability as the yellow zone, and the area with low vulnerability as the green zone. If the resulting values are applied to the map of Changwon, it is possible to derive the land use classification map, as shown in Figure 5.

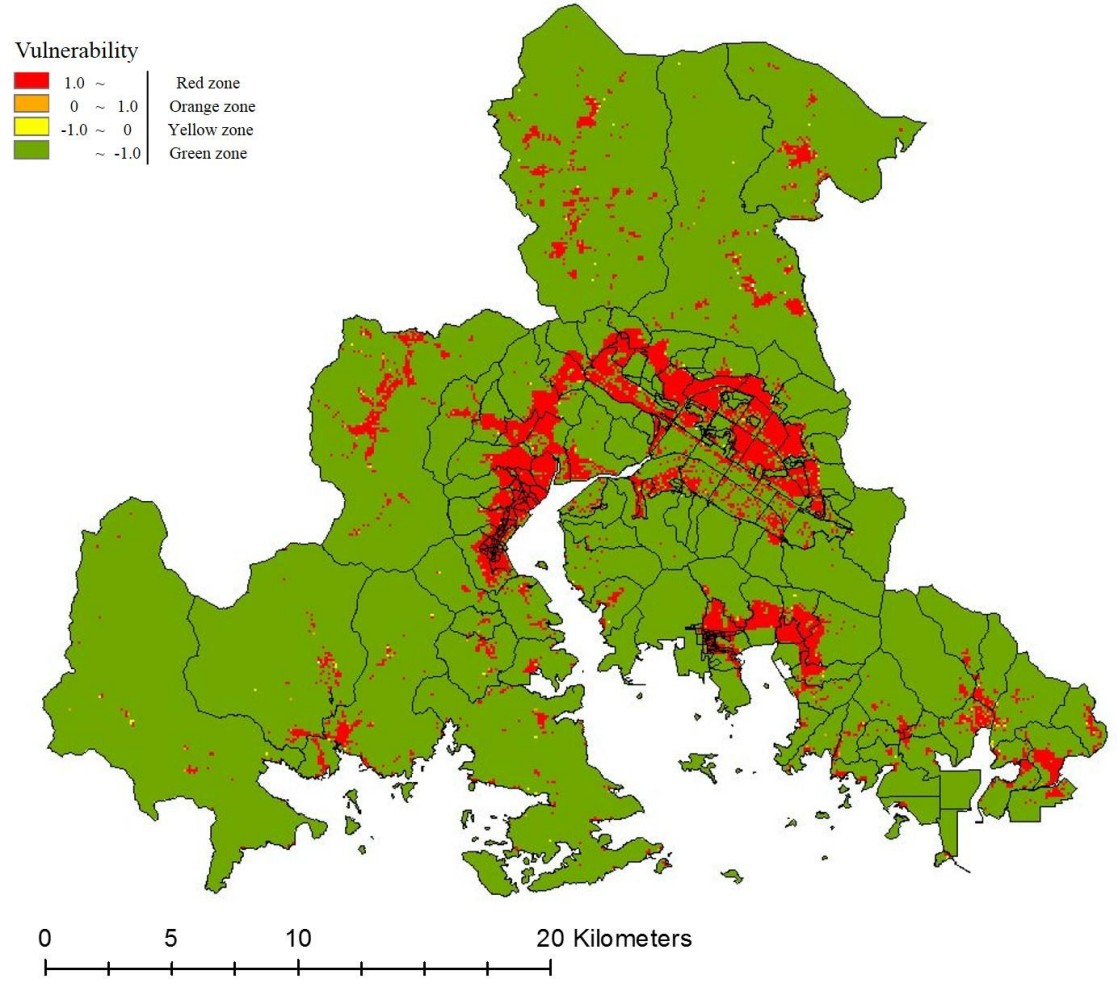

**Figure 5.** Urban flood vulnerability analysis.

*3.2. Analysis of Exposure*

The analysis using HEC-Ras is a typical flood analysis model that conducts a simulation by constructing the river crossing data, the center line of the river, the river bank line, and and using the flood amount and the flood level as the boundary conditions. This analysis process is shown in Figure 6.

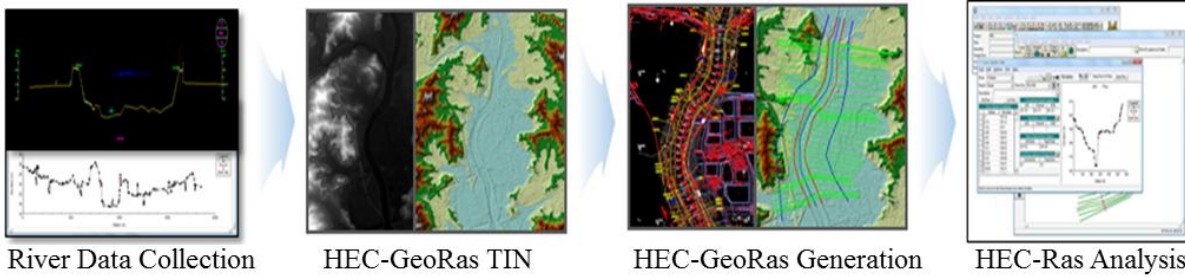

**Figure 6.** Flood depth and flooded area analysis using HEC-Ras.

The urban flood map prepared by using the one-dimensional model HEC-Ras based on the basic environmental factors such as rainfall data and topographical data developed by the Ministry of Interior and Safety was used in this study [45]. The urban flood map is shown in Figure 7 below. The urban flooding area evaluation confirmed that the flood depth was high in the northern area adjacent to the river and reservoir and also high in the areas adjacent to Changwon City Hall, which play key roles in Changwon City and are also considered as the town center.

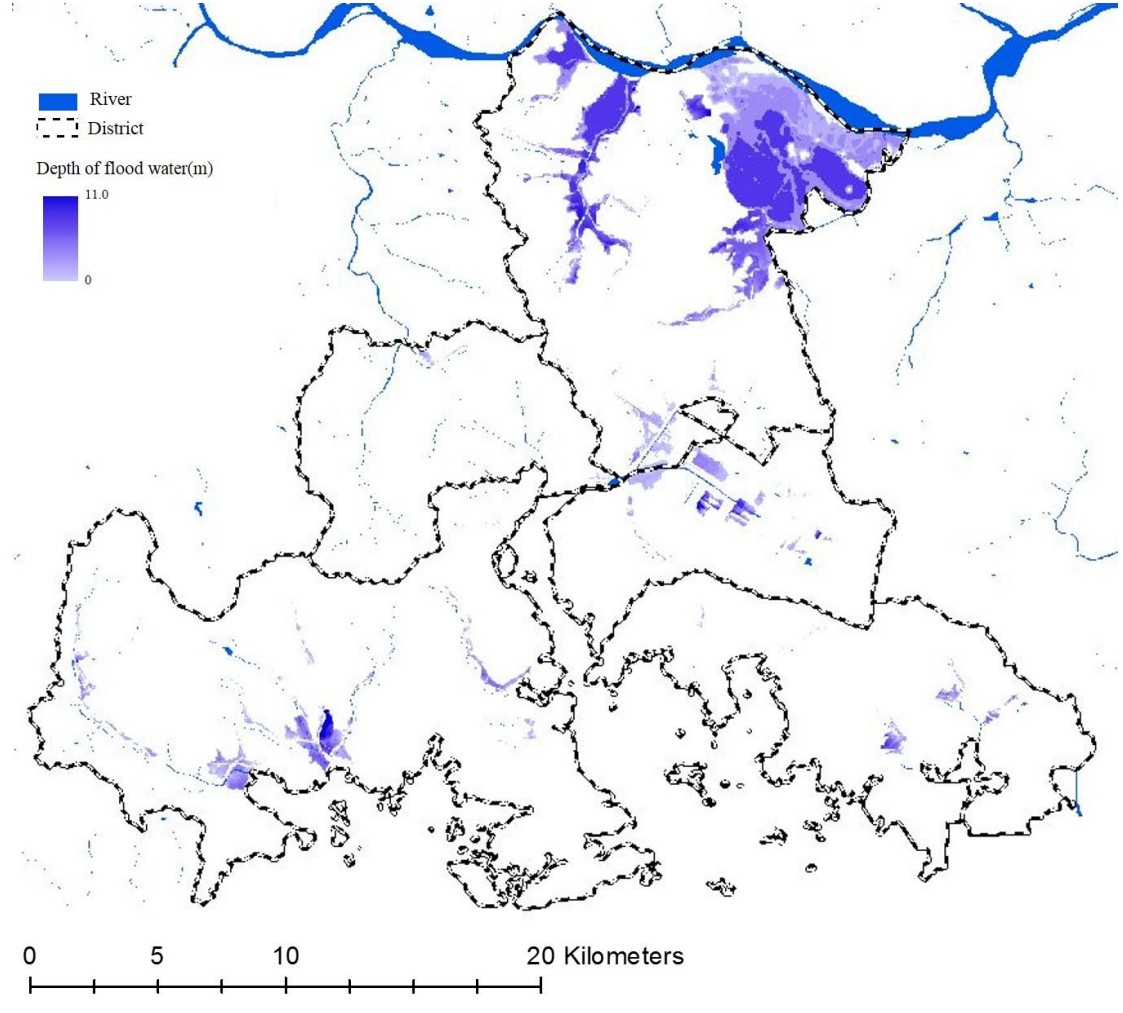

**Figure 7.** Urban flood exposure analysis.

### 3.3. Analysis of Risk

Figure 8 displays the graded urban flood risk by overlapping the non-structural factors, which are the criteria for socioeconomic damages, derived based on the study by Kron [46] and Brooks et al. [47], with the flood simulation analysis map. This map analyzed the flood depth based on the land

use classification map, formed of vulnerability analysis data and data on disaster characteristics (rainfall, topography), including the degree of exposure.

The analysis of urban flood risk confirmed that the red zone was mainly found in the central commercial area, where various infrastructures, including public institutions that play key functions in the city, are located. This appears to be related to the development considering the economy, convenience, and efficiency of the city. The orange zone shows a fan-shaped distribution centering on the red zone, which also demonstrates the tendency of branching out from major functions and socio-economic parts of the city, and by use area, they were classified into distribution commercial areas, including general commercial areas and quasi-residential areas. The distribution of the yellow zone centered on the residential area confirms that it is distributed in the area that was formed a long time ago. It is considered that this is closely related to the aging of buildings. The green zone was mainly distributed in green areas where the environmental values or the environment rather than the value of development land use need to be preserved.

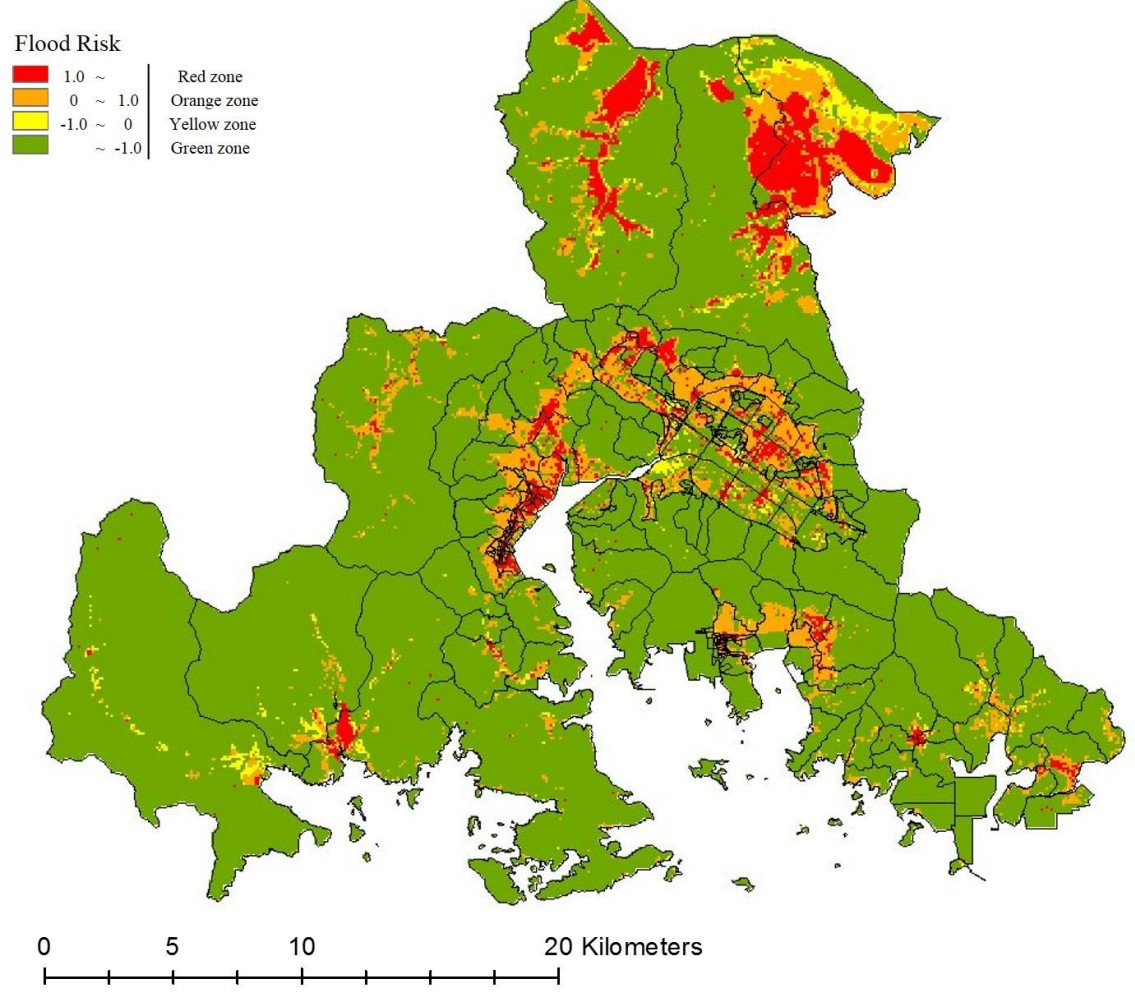

**Figure 8.** Urban flood risk analysis.

*3.4. Overall Analysis Results*

The results of the risk analysis by district are shown in Table 2. The area with the highest urban flood risk and the highest percentage of red zone in urban flood risk was Euichang-gu (13.07%), followed by Seongsan-gu (4.07%), Masan Hoewon-gu (2.68%), Masan Happo-gu (1.87%), and Jinhae-gu (1.78%), in respective order. The orange zone was also distributed more frequently in the window (14.78%) and in Seongsan-gu (12.37%) than in other areas. The distribution of the yellow zone was most prominent in Euichang-gu (3.96%), followed by Seongsan-gu (2.10%) and Masan Happo-gu (1.39%), respectively. The green zone, which is the safe area with the lowest urban flood risk, was most distributed in Masan Happo-gu (92.21%), Jinhae-gu (86.99%), and Masan Hoewon-gu (85.89%).

Masan Happo-gu has been active as the general commercial area since being developed a long time ago centering on Masan Port and served as the transportation hub with railroads in the past, although it is now an abandoned railroad site. The neighborhood commercial area and semi-residential area were formed around the district, and the urban flood risk in these areas appears to be high. The flood risk in the industrial areas is relatively low as they were recently formed sporadically in mountainous areas. Considering the results in terms of building characteristics, a high fuzzy value for construction year across the whole use area indicates that the city was formed long ago and that the aging of buildings is becoming serious. Additionally, the basement is distributed more in the industrial area than other use areas.

In Masan Hoewon-gu, a general commercial area is mainly formed around Masan Station and its neighborhood, with commercial areas, semi-residential areas, and general residential areas around it. The industrial area was extensively formed in the southeastern area in the past and the flood risk is more or less high as the vulnerability of the building materials is high. According to building characteristics, the construction year was higher than other indicators in the same way as Masan Happo-gu, indicating that the buildings are aged. In particular, the officially assessed land price of the neighboring commercial area is much higher than other indicators, and this suggested that there is a great demand for daily necessities and services from the residents of the neighborhood residential area and the area appeared to have been specialized for that purpose.

Seongsan-gu is a comparatively new city area, where the commercial area and residential area have been recently developed, centering on public institutions including the city hall, and the fuzzy value for the materials is low as the buildings are not aged and many of them were constructed recently. The officially assessed land price and floor area ratio are high throughout the use district, particularly in the general commercial area, central commercial area, and semi-residential area, indicating that it is an area with high asset value. The fuzzy value is high, particularly as the highly dense central commercial area is activated, and also as many semi-residential areas incorporate commercial facilities among the residential areas. The reason why the fuzzy value is low in the green area is that it has a low asset value and it is quite unlikely that development of the area would occur in this region.

Euichang-gu is located near Seongsan-gu. Like Seongsan-gu, the commercial areas and residential areas have recently been developed around public institutions, including the provincial government buildings, and the fuzzy value of the city's land price is considerably higher than other indicators, indicating that the asset value of the region is high. The green area demonstrated the lowest fuzzy value like other areas, indicating the lowest vulnerability.

Jinhae-gu has a low asset value and the fuzzy value of the construction year was high throughout the whole use district, indicating that the city has existed for a long time and the buildings are aged compared with other areas. In the Jinhae area, semi-residential areas and general commercial areas showed more or less a higher vulnerability, and fuzzy values of industrial areas and green zones were low.

**Table 2.** Risk for each administrative district.

| | Area (m²) | Green Zone | | Yellow Zone | | Orange Zone | | Red Zone | |
|---|---|---|---|---|---|---|---|---|---|
| | | Cells of Grid Units | Ratio (%) | Cells of Grid Units | Ratio (%) | Cells of Grid Units | Ratio (%) | Cells of Grid Units | Ratio (%) |
| Masan Happo-gu | 239,630,000 | 22096 | 92.21 | 332 | 1.39 | 1087 | 4.54 | 448 | 1.87 |
| Masan Hwewon-gu | 90,840,000 | 7802 | 85.89 | 31 | 0.34 | 1008 | 11.1 | 243 | 2.68 |
| Seongsan-gu | 82,080,000 | 6687 | 81.47 | 172 | 2.1 | 1015 | 12.37 | 334 | 4.07 |
| Euchang-gu | 211,320,000 | 14410 | 68.19 | 836 | 3.96 | 3124 | 14.78 | 2762 | 13.07 |
| Jinhae-gu | 120,410,000 | 10,475 | 86.99 | 81 | 0.67 | 1271 | 10.56 | 214 | 1.78 |

By use district, the ratio of red zones of commercial areas, including central commercial areas and general commercial areas, was higher, as shown in Table 3, and the risk declines in the order of residential areas > industrial areas > green areas.

**Table 3.** Risk for each use district.

| Use District | | Area (m²) | Green Zone | | Yellow Zone | | Orange Zone | | Red Zone | |
|---|---|---|---|---|---|---|---|---|---|---|
| | | | Cells of Grid Units | Ratio (%) | Cells of Grid Units | Ratio (%) | Cells of Grid Units | Ratio (%) | Cells of Grid Units | Ratio (%) |
| Residential Area | Private Residential Area | 11,300,000 | 249 | 22.04 | 18 | 1.59 | 767 | 67.88 | 96 | 8.5 |
| | General Residential Area | 44,000,000 | 1374 | 31.23 | 63 | 1.43 | 2265 | 51.48 | 698 | 15.86 |
| | Semi-residential Area | 2,080,000 | 43 | 20.67 | 3 | 1.44 | 106 | 50.96 | 56 | 26.92 |
| Commercial Area | Central Commercial Area | 1,230,000 | 37 | 30.08 | 0 | 0 | 44 | 35.77 | 42 | 34.15 |
| | General Commercial Area | 6,670,000 | 84 | 12.59 | 6 | 0.9 | 236 | 35.38 | 341 | 51.12 |
| | Neighborhood Commercial Area | 240,000 | 6 | 25 | 1 | 4.17 | 13 | 54.17 | 4 | 16.67 |
| | Distribution Commercial Area | 900,000 | 36 | 40 | 4 | 4.44 | 33 | 36.67 | 17 | 18.89 |
| Industrial Area | | 34,300,000 | 2269 | 66.15 | 138 | 4.02 | 773 | 22.54 | 250 | 7.29 |
| Green Area | | 356,260,000 | 33,270 | 93.39 | 443 | 1.24 | 1489 | 4.18 | 424 | 1.19 |

These results imply that the precautionary measures for flooded areas should primarily focus on the central commercial areas among the use district of the urban areas. In addition, the green zones have the least risk, as shown in the analysis results, and the potential to reduce or minimize the flood risk and may demonstrate more flexibility than the structural measures in various aspects considering the uncertainty of climate change. Moreover, the green zones are more effective than aged structural systems (sewage, storage facilities) in the long term, and not only reduce the rainwater runoff rate, but also reduce water pollution, which makes them one of the most useful resources, while the structural (sewer) system generates pollution and increases water pollution. Therefore, the results confirmed the fact that the flood risk could be reduced by arranging green zones appropriately.

In order to apply measures in terms of urban planning to mitigate urban flood damages, it is necessary to establish an objective and reasonable method of setting dangerous areas, such as disaster prevention areas, and this study emphasizes that measures for disaster prevention shall be established by dividing the results of the urban flood risk analysis into 100 m × 100 m geographic grids and primarily considering the areas with a relatively high ratio of red zones.

## 4. Conclusion

The development and expansion of the city have brought about a variety of environmental problems, and the disaster in the city is gradually becoming larger and diversified due to the influence of climate change. Under this circumstance, the importance of urban disaster prevention has been emphasized, and this study focused on flooding, which accounts for a significant portion of disasters related to climate change. This study intended to contribute to the reduction and minimization of flood damage in the case of heavy rainfall in urban areas, which are concentrated with population and major facilities, by increasing the urban spatial efficiency through the grading

of flood risk. This study developed and applied a flood risk assessment model to Changwon City and obtained the results described below.

First, as an assessment model, an objective and scientific evaluation model which reflects urban spatial concepts and characteristics of buildings was developed by applying an urban flood risk assessment model. Additionally, the non-structural characteristics were identified and the land use and vulnerability of building units were analyzed to minimize damage from natural disasters as a long-term measure. This study intended to construct an urban spatial model to primarily apply to less vulnerable and risky areas and suggested a new paradigm for the integrated study of 'environment + city' rather than individual planning or the study of environmental planning and urban planning. The developed model provides urban planners with a flood risk map when developing flood risk management strategies to ensure that a strong decision is made on flood management options such as land use, which represents a major challenge for flood risk, and disaster management challenges faced by climate change. Therefore, the ability of urban planners to make such decisions is important in reducing the social, economic, and physical (infrastructure) impact of flood damage.

Second, a set of indicators with non-structural characteristics that are related to land use and building characteristics were derived to assess vulnerability. The priority for each 100 m × 100 m geographic grid for flood damage was determined based on the fuzzy analytical values for each indicator: officially assessed land price, floor area ratio, underground area, decline of building, and material of building. When the same area was flooded, the area with the largest flood damage in a social and economic sense, which is the area with high vulnerability, was the area with the highest development density, including the central commercial area. The areas that had key functions in the city in the past, as the old town center, although currently declining, demonstrated high vulnerability.

Third, an urban flooding map was constructed by using the HEC-Ras model based on the basic environmental factors, such as rainfall data and topographic data, which were developed by the Ministry of the Interior and Safety. The results of urban flooding area evaluation confirmed that the flood depth of the northern area adjacent to the river and reservoir is high. The flood depth was also high in the areas adjacent to Changwon City Hall, which play key roles in Changwon City, the target area, and which are also considered as the town center.

Fourth, the flood risk was analyzed by overlapping the results of vulnerability analysis and exposure analysis. As a result of analyzing the risk of urban flooding by four grades: green, yellow, orange, and red zones, it was confirmed that red zones were formed centering on the central commercial areas, where a variety of infrastructures, including public institutions with key functions in the city, were developed. This is related to the development of the economic, convenience, and efficient aspects of the city. The orange zone has a fan-shaped distribution centering on the red zone, which also shows the tendency of branching out from major functions and socio-economic parts of the city, and by using districts, they were classified into distribution commercial areas, including general commercial areas and semi-residential areas. The distribution of the yellow zone centered on the residential area, confirming that it is distributed in the area that was formed a long time ago. It is deemed that this is closely related to the aging of the buildings. The green zone was mainly distributed in green areas, where the environmental values or the environment rather than the value of development land use need to be preserved.

Lastly, the analysis of the urban flood risk in each administrative district showed that Euichang-gu (13.07%) had the highest urban flood risk with the highest percentage of red zone in Changwon, followed by Seongsan-gu (4.07%), Masan Hoewon-gu (2.68%), Masan Happo-gu (1.87%), and Jinhae-gu (1.78%), in respective order. This was attributed to the generally high property values and building density in the commercial areas of Seongsan-gu and Euichang-gu, which represent the central region for new town development. In other words, the land use plan should be established by allocating the green zone with the lowest urban flood risk and other use districts appropriately and disaster prevention facilities and space facilities that can reduce flooding should be allocated appropriately to prepare measures primarily for the areas with high risk. However, the difference in the analysis results of the five districts, which differed from each other, indicates that the degree of damage from

disasters may vary according to the local environment and characteristics, newly developed town areas, the decline of the old town center and geographical location.

**Author Contributions:** K.P. (Kiyong Park) conceptualized the research, performed the formal analysis, and wrote the first draft of the paper. M.H.L. (Man-Hyung Lee) provided feedback on the research approach, and reviewed the first draft of the paper. All authors revised the paper and agreed on the final version of the paper.

**Acknowledgements:** This paper was financially supported by Ministry of the Interior and Safety as "Human resource development Project in Disaster management".

**Conflicts of Interest:** The authors declare no conflict of interest.

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
