# Peer review of "The Development and Application of the Urban Flood Risk Assessment Model for Reflecting upon Urban Planning Elements"

_water, doi:10.3390/w11050920_

Round 1
Reviewer 1 Report
This paper presents an urban flood risk assessment model in order to reduce potential fatalities and damage in the urban areas of Changwon City, Korea. Apart from hazard analysis, this approach includes the damage assessment by taking several urban characteristics (e.g. land price, floor area ration, underground area, …) into account. In addition, the framework uses fuzzy methods for analyzing the vulnerability. This research is practical for government and commercial industries (e.g. insurance companies). However, several points must be clarified before this draft can be taken into consideration for publication. The main concerns are:
1. The infrastructure of the system presented in this paper is following the typical flood risk estimation process – water level-inundation map-flood damage. I won’t treat it as a new framework. It’s more about the implementation of the typical process for special case study.
2. Considering the differences between standard risk estimation processes for fluvial flooding and pluvial flooding, the methods presented are useful for river-floods. Urban flooding by heavy rainfall-induced flash floods cannot be considered.
3. An introduction is normally following the steps of (1) relevance, (2) known methods, (3) unknown methods, (4) objective / aim of the paper, (5) results. However, the introduction presented is just showing up (1) and (2). Furthermore, the problem presented in line 67-75 are no new developments, at least in global perspective.
4. The method/process described in chapter 2.4 is confusing for the reader. The authors should explain in detail the steps of the applied risk analysis.
5. Furthermore, the chosen spatial resolution of 100 x 100 m is insufficient for urban flooding analysis including the hazard phenomena of pluvial flooding.
Still, I think there are values to publish such a study if the authors could present the experiences during the implementation of the risk assessment model. Try to interpret your system in the context of a standard flood damage assessment process (frequency-discharge-stage-damage, as presented on page 34 of the Expected Annual Flood Damage Computation manual from the US arm crop; https://www.hec.usace.army.mil/publications/ComputerProgramDocumentation/CPD-30.pdf). Furthermore, a detailed discussion is missing to evaluate the developed risk assessment model in the context of existing models.
More professional editing is needed to correct some major grammatical errors.
A set of technical issues and comments for the paper are provided here:
· Line 38: What do you mean with “large-scale flash floods”?
· Line 79-80: Please define “in the past Since a very long time humans are “fighting” against flood hazards by flood protection measures using dams and dikes. Also
· Line 91-93: “Reducing the area for urban land use would naturally reduce the resilience of disaster damages..”. Those statements are following their logic of worldwide accepted statements.
· Line 95-96:
· Line 131-141: There are much more risk-definitions.
· Line 177: The content of Figure 2 is hard to understand. What is the primary objective?
· Line 184: Which risk mapping technique?
· Line 207: Figure 4 is missing.
· Line 225: The fuzzy methodology used should be explained in detail in chapter 2.
· Line 265: The location is missing here.
· Line 267-270: Complex sentence. What is the spatial and temporal resolution of the hazard analysis?
· Line 273-278: In my point of view a 1D-only simulation and analysis of urban flooding processes are not enough in order to assess the hazard. In addition, a hydrodynamic 2D-model is needed.
Author Response
Dear. Reviewer
Thank you so much for your comments and suggestions.
I greatly appreciate your efforts and helpful comments in reviewing my paper.

Reviewer 2 Report
This manuscript presents a series of data analyses on urban flooding risks in different areas. The results give some predictions for facilities building. The English language needs to be edited. The manuscript presents some efforts on data analyses. I recommend for publication with major revision. Please see the details in the attachment.

Author Response

(The authors gave the same response as above.)

Reviewer 3 Report
The authors present a study on the development and application of the urban flood risk assessment model reflected upon urban planning element. The paper is interesting reading and is well-written and well-structured. In general, I am happy with the paper. However, I do have some minor concerns about the paper that need to be addressed:
The introduction lacks a clearly stated research objective, but also a description of the geographical scope. This leaves the reader without any scope. Please describe.
Methods: the authors do not clearly describe how data(sets) have been used/curated for their analysis. Please describe.
Results: p8, line 258-259, what are the weight quotients?
Results: How are uncertainties taken into account in the analysis?
Conclusion: p13, line 382-283, I am not yet convinced of the authors' claim that the study intended to construct an urban spatial modelling to primarily less vulnerable and risky areas and suggested a new paradigm for the integrated study of 'environment 383 + city' rather than individual planning or study of environmental planning and urban planning. Please describe how the method developed could inform/support urban planners in developing flood risk management strategies? See e.g.,
Buijs, J.M., Boelens, L., Bormann, H., Restemeyer, B., Terpstra, T., van der Voorn, T., (2018). Adaptive planning for flood resilient areas: dealing with complexity in decision-making about multilayered flood risk management. Conference paper for the 16th AESOP meeting on Adaptive Planning for Spatial Transformation (2018) in Groningen, the Netherlands.
Hegger, D. L. T., P. P. J. Driessen, C. Dieperink, M. Wiering, G. T. Raadgever, and H. F. M. W. Van Rijswick (2014), Assessing stability and dynamics in flood risk governance: An empirically illustrated research approach, Water Resour.Manag., 28, 4127–4142, doi:10.1007/s11269 014-0732-x.
Please check minor details in the attachment.

Author Response

(The authors gave the same response as above.)

Round 2
Reviewer 2 Report
The manuscript is presented well and I recommend it for publication.
Author Response
Thank you again.
